# Mariana-type ophiolites constrain the establishment of modern plate tectonic regime during Gondwana assembly

Jinlong Yao [1✉], Peter A. Cawood [2], Guochun Zhao[1,3✉], Yigui Han[1], Xiaoping Xia [4], Qian Liu[3] & Peng Wang[3]

Initiation of Mariana-type oceanic subduction zones requires rheologically strong oceanic lithosphere, which developed through secular cooling of Earth's mantle. Here, we report a 518 Ma Mariana-type subduction initiation ophiolite from northern Tibet, which, along with compilation of similar ophiolites through Earth history, argues for the establishment of the modern plate tectonic regime by the early Cambrian. The ophiolite was formed during the subduction initiation of the Proto-Tethys Ocean that coincided with slab roll-back along the southern and western Gondwana margins at ca. 530-520 Ma. This global tectonic reorganization and the establishment of modern plate tectonic regime was likely controlled by secular cooling of the Earth, and facilitated by enhanced lubrication of subduction zones by sediments derived from widespread surface erosion of the extensive mountain ranges formed during Gondwana assembly. This time also corresponds to extreme events recorded in climate and surface proxies that herald formation of the contemporary Earth.

[1] State Key Laboratory of Continental Dynamics, Department of Geology, Northwest University, Northern Taibai Street 229, Xi'an 710069, China. [2] School of Earth, Atmosphere & Environment, Monash University, Melbourne, VIC 3800, Australia. [3] Department of Earth Sciences, The University of Hong Kong, Pokfulam Road, Hong Kong, Hong Kong SAR. [4] State Key Laboratory of Isotope Geochemistry, Guangzhou Institute of Geochemistry, Chinese Academy of Sciences, Guangzhou 510640, China. ✉email: yaojinlong@nwu.edu.cn; gzhao@hku.hk

The commencement and evolution of the plate tectonic regime on Earth is linked to secular mantle cooling and associated increasing lithospheric strength[1–5]. The latter leading to lithospheric thickening and increased surface erosion and sediment influx to trenches, resulting in subduction lubrication and reduction in the average friction coefficient (by more than one order of magnitude) along the subduction zone interface, and increasing subduction velocity and sustaining subduction[6]. Some form of plate tectonics, possibly being plume-induced retreating subduction type, or warm subduction type or squishy lid type, is generally inferred to have initiated in the late Archean, as mantle potential temperature decreased and lithosphere strength increased[4,6–9]; (Fig. 1). In the Paleoproterozoic, after the Huronian glaciation, a plate tectonic regime reached whole plate scale, but may have been followed by a reversion to a non-plate tectonic and nearly static single-lid episode during Earth's middle age[3,10,11]. In the Neoproterozoic-Cambrian, during Gondwana assembly[12–16], as mantle potential temperature continued to decrease below $\Delta T = 80–100\ °C$ (relative to present value) and lithosphere strength further increased[17], the modern

**Fig. 1 Stages of geodynamic regimes with respect to ages of Mariana-type subduction initiation ophiolites and evolution of Earth's geological and surficial proxies. a** The thermal gradient vs. metamorphic ages of three main types of granulite facies metamorphism[1]. **b** The inferred mantle potential temperature[5], thermobaric ratio curve[21]. **c** The running mean of initial εHf in detrital zircons and the running mean of zircon δ18O normalized to average sediment[26], the ages of ancient and modern passive margins[74], the normalized seawater 87Sr/86Sr curve[75]. **d** The changing oxygen levels within the atmosphere relative to the present atmospheric level[76], and the evolution of life within the biosphere[77]. Gray shaded bar corresponds to the Phanerozoic time.

plate tectonic regime was established and marked by stable plate subduction to deep mantle and high P/T UHP metamorphic assemblages[17–22]; (Fig. 1), and the extensive mountain ranges and resultant sediment erosion[23,24]. The Neoproterozoic-Cambrian period also recorded a shift to environs that are more similar to present Earth (Fig. 1)[23–26]. However, the feedbacks between the establishment of this modern tectonic regime and surface environs remain poorly understood.

Subduction initiation is a key component of the plate tectonic paradigm and exerts a major control on the modern Earth system through exchange between surficial and solid Earth reservoirs[27,28]. Key factors in the initiation of a stable modern, Mariana-type (also referred to as Izu–Bonin–Mariana (IBM) type), intra-oceanic subduction zone includes high slab strength, weak zones of focused stress, and mantle wedge hydration[2]. Internal vertical forces are geodynamic necessities to produce typical Mariana-type subduction initiation magmatic products, along with their temporal and spatial distributions[28]. This setting is characterized by the formation of a proto-arc ophiolite (also referred to as a subduction initiation ophiolite), subsequently preserved in the forearc of the resultant convergent plate margin, and contains boninites, basalts, gabbroic rocks, and mantle peridotites[29,30]. The igneous sequences are derived from mantle sources that evolved from the combined effects of melt depletion and subduction-related metasomatism, and form within 7–10 Ma of subduction initiation[2,30–32]. Such a subduction initiation event operates at a whole plate scale (>1000 km) and forms a new destructive zone[28,33]. Ophiolites that form at the initiation of subduction are, however, rarely preserved[31]. Typical examples include the IBM ophiolite[30,33,34] and the late Cretaceous Tethyan ophiolites[32,35].

The Tarim Block lies between the Central Asian Orogenic Belt and the Tibetan Plateau (Fig. 2a). The block is largely covered by desert but Archean-Neoproterozoic rocks are exposed around its margins[36–38], and its southern margin is bounded by the early Paleozoic Kunlun and Altyn orogenic belts (Fig. 2a)[39–45]. The Altyn Belt records multiple orogenic cycles, including Archean-Paleoproterozoic and the latest Mesoproterozoic-Neoproterozoic events that were extensively overprinted in the early Paleozoic[19,38,41–49]. Its northern boundary is the northern Altyn fault zone (Fig. 2a), but its southern margin is less well constrained and is offset by the Cenozoic reactivation of the Altyn fault zone associated with India-Asia collision[40,43]. The Altyn Belt is divisible into the Archean north Altyn terrane and the late Mesoproterozoic-early Paleozoic central and south Altyn terranes, along with two early Paleozoic ophiolite belts (the Hongliugou-Lapeiquan and southern Altyn ophiolitic belts) (Fig. 2a). The Hongliugou-Lapeiquan belt delineates the boundary between the north and central Altyn terranes, whereas the southern Altyn ophiolitic belt is located at the southeastern margin of the South Altyn Terrane (Fig. 2a). Detailed regional geology can be found in the supplementary information. The Munabulake ophiolite occurs as a tectonic block within the South Altyn Terrane, and is extensively sheared and deformed[50].

In this work, we document the ca. 518 Ma Munabulake ophiolite from the Southern Altyn Terrane (also referred to as the Southern Altyn HP-UHP belt) in the southeastern margin of the Tarim Block, northern Tibet. Ophiolite stratigraphy, field relations, ages, zircon Hf-O isotopes and $H_2O$ compositions, along with whole-rock and mineral compositions argue for formation during intra-oceanic subduction initiation. This ophiolite was formed during initiation of Mariana-type oceanic subduction, which we link to other subduction initiation ophiolites and argue for high oceanic slab strength at least since this time. The Munabulake ophiolite is also the first record of the subduction initiation of the Proto-Tethys Ocean, which lay outboard of

northern Gondwana[14,51,52]. We place the timing of subduction initiation in the Proto-Tethys within a broader framework involving tectonic re-organization associated with the establishment of modern plate tectonic regime during Gondwana assembly at around 530–520 Ma, and linking this to changes in climate and biosphere.

## Results

**Stratigraphy and sampling of the Munabulake ophiolite in the South Altyn Terrane.** The Munabulake ophiolite is thrust upon other units of the latest Mesoproterozoic-early Paleozoic Altyn complex along its northern margin, whereas on its southwestern margin, a NW-SE directed sinistral strike-slip ductile shear zone defines the boundary with the remainder of the Altyn Complex (Fig. 2b, c). Original contacts between units within the complex remain unclear as they experienced long-term high-grade metamorphism and extensive deformation in the early Paleozoic, at ca. 500–420 Ma, and in the early Mesozoic[19,40,43].

From northeast to southwest, and corresponding with the progression from base to top, the Munabulake ophiolite is composed of sheared serpentinite along the basal thrust, serpentinized dunite-harzburgite, pyroxene peridotite, olivine pyroxenite, gabbro, and meta-basaltic and meta-intermediate igneous suites, along with blocks of marble (Fig. 2b, S1a, S1b, S1c, S1d). Fine-grained siliceous rocks ranging in size from centimeters to meters thick are interlayered with the basaltic and intermediate igneous suites (Fig. S1c, S1e). They are interpreted to represent recrystallized chert and are indicative of a deep-sea marine environment. Ultramafic blocks of serpentinized dunite and pyroxene peridotite, up to 7 km thick, are exposed in the northwest segment of the ophiolite (Fig. 2b, 2c, S1a, S1b). An olivine pyroxenite block, ~2 km thick, is also sandwiched between blocks of serpentinized dunite and pyroxene peridotite, whereas a harzburgite block is sandwiched between blocks of serpentinized dunite (Fig. 2b). The ultramafic rocks, generally serpentinized, are interpreted as the lower mantle components of the ophiolite. A structural block that consists mainly of meta-gabbro and meta-basaltic-intermediate suites occurs within the ductile shear zone along the southwest margin of the ophiolite (Fig. 2b, S1c). Pillow basalt, along with deep-sea turbidites form the upper crustal components of the ophiolite stratigraphy[50], but diabase dikes are less developed. These suites form the crustal components of the ophiolite. In summary, the overall lithological assemblage is consistent with a disrupted ophiolite, and occurs within gneisses of the Altyn Complex. Four peridotite samples and 17 basaltic-intermediate samples are selected from the ophiolite for zircon U-Pb ages, Hf-O isotopes, and $H_2O$ contents, along with whole-rock and mineral compositions. Detailed regional geology and sample descriptions are in the supporting information.

**Mineral isotope and chemistry.** Forty-two analyses on zircon grains from the meta-gabbro sample 17SAT13-2 yielded a concordant $^{206}Pb/^{238}U$ age range of 543–506 Ma and a weighted mean age of 518 ± 2 Ma (MSWD = 0.94, $n = 40$) (Fig. S2), which display present day $^{176}Hf/^{177}Hf$ ratios of 0.282792–0.282863, equivalent to εHf(t) values of 11.1–13.6 and Hf model ages of 566–670 Ma (Fig. 3a). The grains also have δ$^{18}$O compositions ranging from +2.69 to +5.7‰ (Fig. 3b) that are either comparable with, or lower than, mantle zircons[53]. Thus, zircon Hf-O isotopes argue for the crystallization of zircon grains from mantle-derived mafic magmas. In addition, $H_2O$-in-zircon contents of these grains range between 109 and 1339 ppmw, with two peaks at 260 and 520 ppmw (Fig. 3b). Less serpentinized harzburgite samples 17SAT22-1 and 18SAT41-4 and olivine pyroxenite sample 18SAT41-5 were selected for olivine and spinel

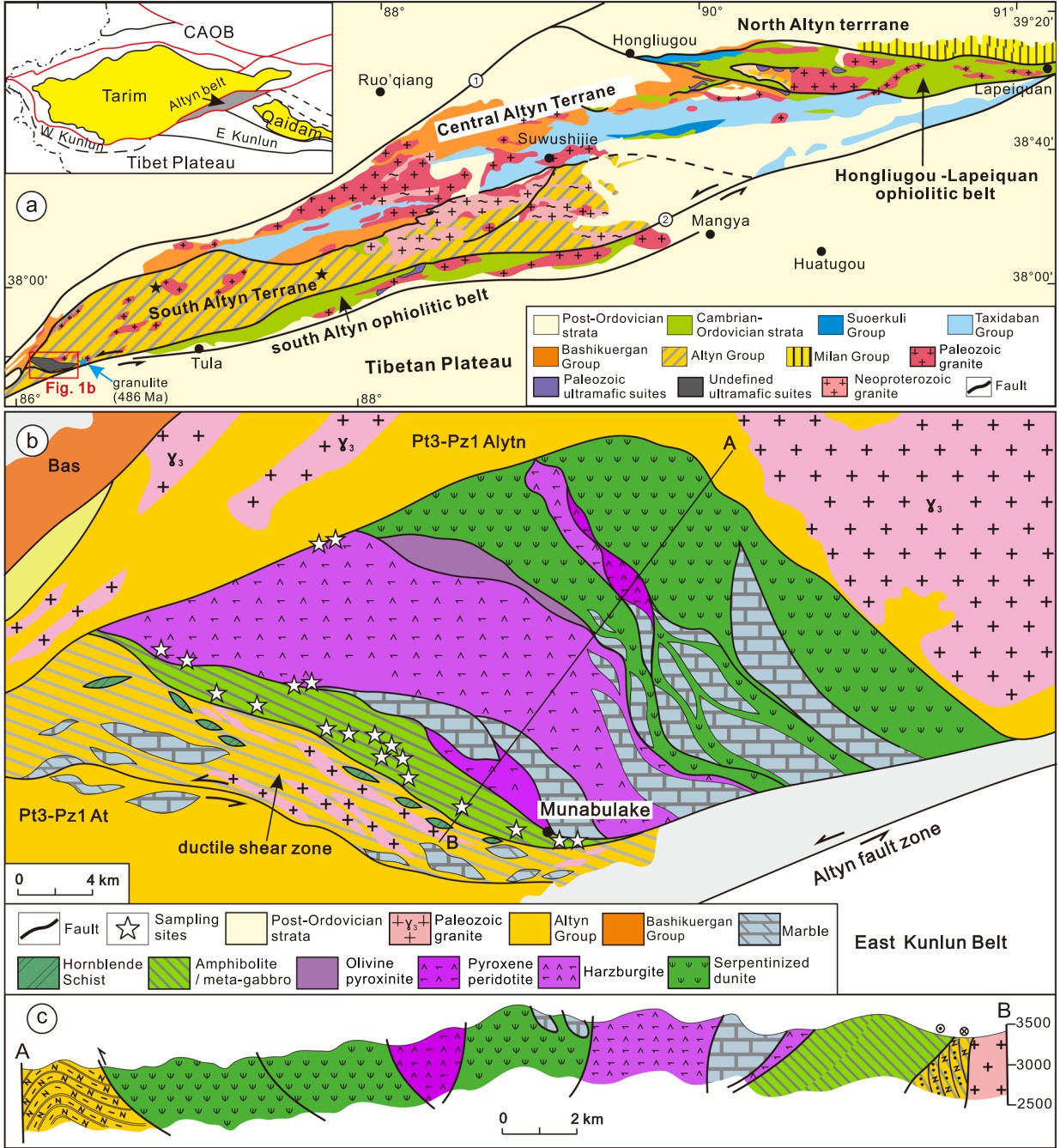

**Fig. 2 Geological sketch map and cross section of the Altyn belt and Munabulake area. a** Geological sketch map of the Altyn Belt. **b** Geological map of the Munabulake ophiolite. **c** Regional cross section of the Munabulake ophiolite.

compositional analyses (Fig. S1f), particularly because peridotite spinels are resistant to secondary alteration processes and preserve a record of peridotite formation[54,55]. The olivines in the pyroxenite have Fo values of 90.9–91.7 and NiO contents of 0.256–0.524 wt%, whereas in the harzburgite they show Fo values between 89.9 and 91.9, and NiO contents of 0.25 and 0.46 wt% (Fig. 4). The Cr# values of spinels in the olivine pyroxenite (26–77) are lower than depleted harzburgite (71–84) and display a linear trend of Cr# vs. Mg# values (Fig. 4). Analytic procedures and results are in the supporting information.

**Chemical duality of the Munabulake ophiolite: progressive evolution from MORB to SSZ during intra-oceanic subduction initiation.** Chemical composition analysis of this study focuses

on the immobile elements and element ratios, considering the potential for the mobilization of large ionic radius elements during metamorphism and alteration. The mantle members of the Munabulake ophiolite consist of residual dunite, harzburgite, and olivine pyroxenite (Fig. 2b), and display low $TiO_2$ and $Al_2O_3$, but high MgO with Mg# of 90–92. The negatively correlated Cr# vs. Mg# values of spinels within harzburgite and olivine pyroxenite samples suggest that the investigated rocks represent residual mantle, with the olivine pyroxenite related to low degrees of partial melting and melt extraction and the harzburgite to higher degrees (Fig. 4b). The correlation between Mg# values of olivines and the Cr# values of spinels is conformable with the olivine–spinel mantle array (Fig. 4c), also indicating the peridotite samples are residues of various degrees of melt extraction. The

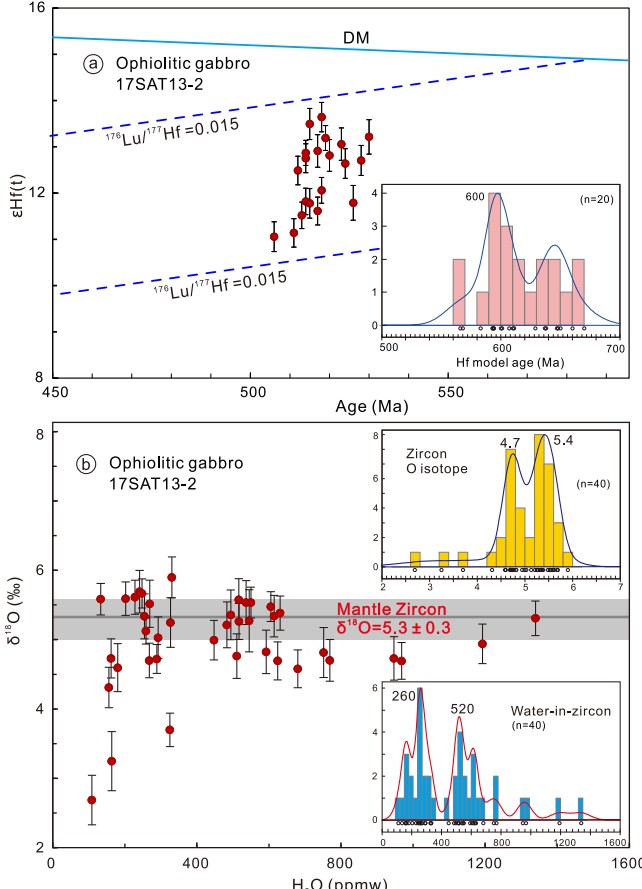

**Fig. 3 Zircon Hf-O isotopes and water contents of the ophiolitic gabbro. a** Hf isotopes. **b** $\delta^{18}O$ vs. $H_2O$ content for zircons from 17SAT13-2. Error bars are 2 SE.

spinel $TiO_2$ vs. Cr# trend suggests a similar magma process (Fig. 4d). The residue ultramafic samples have higher Mg# (>90) values than primitive mantle, and also display U–shaped REE patterns and that are indicative of melt introduction[34] (Fig. S3), consistent with volatile-rich conditions in a supra-subduction zone setting. Euhedral, possibly secondary, tremolite grains are also observed in the peridotite (Fig. S1g), arguing for high fluid metasomatism, as are their varied Zr/Hf ratios. The magma evolution thus reflects progressive source depletion coupled with increasing melt or fluid metasomatism. Therefore, compositions of ultramafic rocks and spinel-olivine minerals are consistent with progressive chemical evolution from abyssal to arc-related peridotite.

The meta-basaltic and meta-intermediate samples from the Munabulake ophiolite, mostly tholeiitic, have low $TiO_2$ (<1 wt%), mostly flat REE patterns (Figs. S3 and S4), along with obvious depletions of Nb and Ta, and very minor Ti depletions for some samples. LREE depleted patterns are observed in two samples. In addition, the mafic-intermediate samples are characterized by elevated Th in the Nb/Yb vs. Th/Yb diagram (Fig. S5a), with most of the samples lie above the mantle array, indicating Th enrichment in their mantle source. This observation is consistent with melting of a mantle wedge that was metasomatized by fluid or melts likely derived from subducted slab sediment. Moreover, high U and varied Th also favor the involvement of hydrous fluids from the altered oceanic crust (Fig. S6a, S6b). This chemical evolution trend indicates increasing source metasomatism by slab-derived material (elevating Th, U, and some LREE). The crustal member samples plot in the SSZ type ophiolite field in

various tectonic discrimination diagrams, including Ta/Yb vs. Th/Yb and Ti vs. V (Fig. S5). Their lower $TiO_2$ (<1.25 wt%), Zr/Y (<3), Nb/La (<0.5), and Ta/Yb (<0.1), along with higher Th/Nb (>0.2) and Th/Yb (>0.1) (Table S2), are comparable to those of oceanic arc basalts[56]. Although these compositional features indicate metasomatism by a subducted slab, their flat REE patterns argue against a normal mature island arc setting. In addition, even compared to lavas of nascent oceanic island arcs, such as Saipan-Rota-Guam[57], the mafic-intermediate samples lack marked Eu and Ti anomalies (Fig. S3b). These traits indicate no plagioclase fractionation and very minor fractionation of Ti-bearing minerals. These signatures suggest that no normal arc or felsic crust had formed at the time of ophiolite formation. In addition, the lower Zr/Y values of these igneous rocks argue for derivation from a depleted mantle source (Fig. S6c), whereas Hf/Nb vs. Zr/Nb suggests magma enrichment by subducted fluids[58]; (Fig. S6d). The overall chemical signatures and evolution trends are comparable to crustal members of the IBM and Tethyan ophiolites (Figs. S3, S4, S5)[30–32]; consistent with the observations from Munabulake mantle end-members. More importantly, zircon Hf isotopes display negative correlation with their ages (Fig. 3a), indicative of the progressive formation of less juvenile magma component during subduction initiation. In addition, $H_2O$ contents in mantle zircons within the ophiolitic gabbro indicate possible similar processes, with two populations at 160–320 and 480–640 ppmw (Fig. 3b). The low $H_2O$ content population is comparable to that observed from Atlantic MORB zircons[59], whereas the higher water content population indicates possible progressive magma hydration or evolution due to the formation of a subduction zone. In addition to the mantle zircons in the ophiolite, zircons with lower $\delta^{18}O$ isotopes (+2.69 to +5.0‰, peaked at 4.7‰) are also observed, which are comparable to altered lower oceanic crust[60]. All the observations suggest linkages between progressive source depletion and metasomatism due to slab-derived fluids, which is typical of a subduction initiation ophiolite and is consistent with observations from mantle end-members of the Munabulake ophiolite. However, it is noteworthy that the mafic-intermediate units occur within a ductile shear zone and were subjected to multi-stage higher grade metamorphism and shearing. Thus, detailed reconstruction of overall chemo-stratigraphy is difficult.

## Discussion

**Direct record of oceanic subduction initiation at ca. 518 Ma.** The south Altyn margin is characterized by the following: (1) Munabulake ophiolite dated at 518 Ma; (2) the 508–475 Ma UHP-HP metamorphism[19]; (3) ca. 508–505 Ma arc-related magmatism at a few localities to the north of the Munabulake ophiolite (author unpublished data); (4) ca. 510–500 Ma MORB type mafic-ultramafic rocks and ca. 503 Ma adakite-diorite (Fig. S7)[48]; and reference therein; (5) ca. 517 Ma oceanic type adakite, along with calc-alkaline granitoids dated at ca. 503–497 Ma occurring in various localities (Fig. S7); e.g[45–48]; and, (6) sinistral shearing along the fault zone at southern margin of the Munabulake ophiolite sometime after ca. 235 Ma (Fig. 2b)[40,43], this study.

The overall ages and field relations indicate that the 518 Ma Munabulake ophiolite is the oldest oceanic succession in the Southern Altyn, followed by establishment of an intra-oceanic arc system lasting some 20 Ma, at least until ca. 500 Ma, as is inferred from the 510–500 Ma MORB type mafic-ultramafic suites and ca. 503–497 Ma arc type granitoids and adakite-diorite. Thus, the crustal and mantle members of the Munabulake ophiolite, along with available data across the south Altyn, are consistent with the subduction initiation signature of the 518 Ma ophiolite, which we

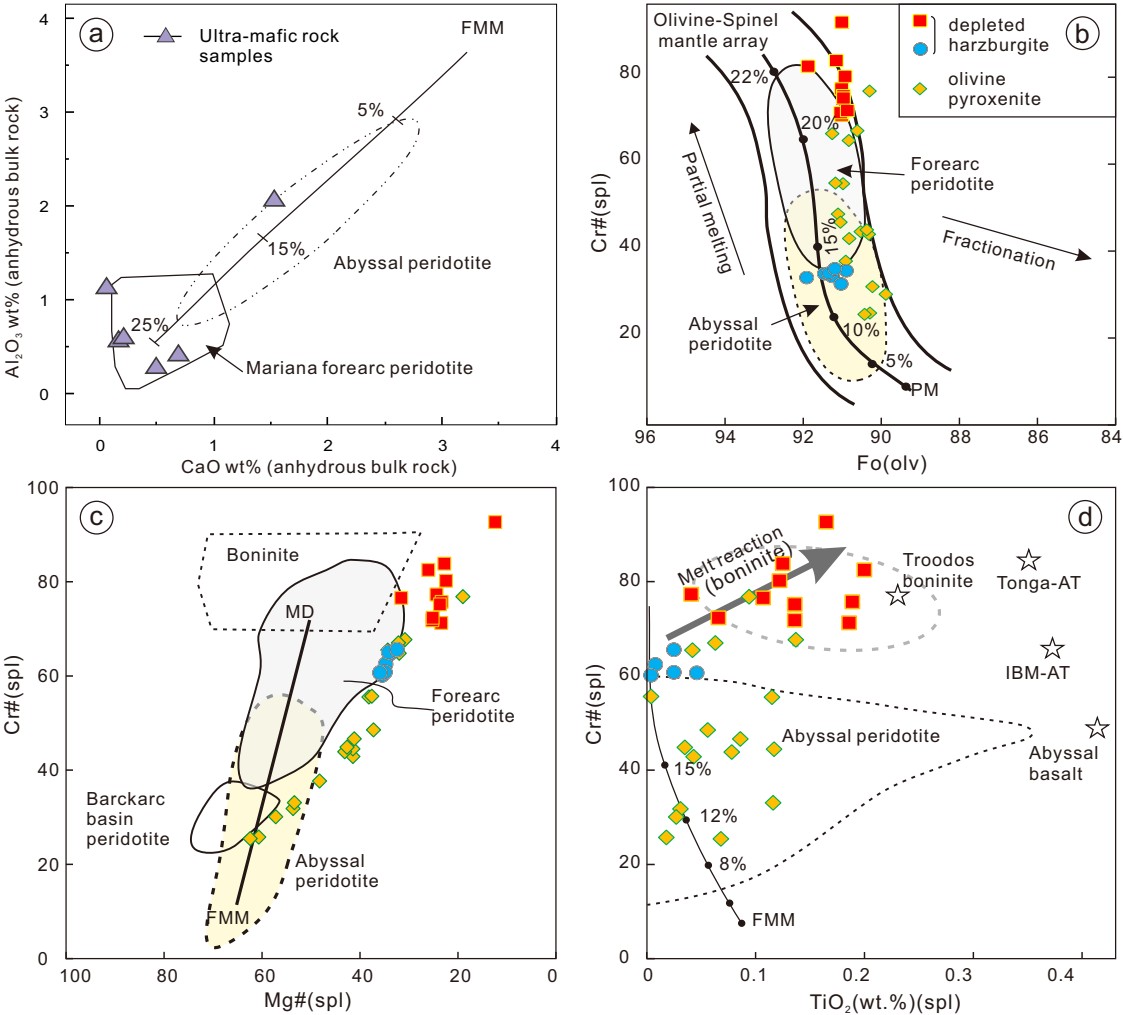

**Fig. 4 Tectonic discrimination diagrams for the ophiolitic peridotites. a** Bulk-rock abundances of $Al_2O_3$ vs. CaO (volatile free, normalized to 100% total), showing how melting depletes peridotites in these elements. **b** Cr# (Cr/Cr + Al) vs. Mg# (Mg/Mg + $Fe^{+2}$) diagram for composition of spinels in peridotite. **c** Compositional relationship between Cr# of spinel and Fo [100 Mg/(Mg + $Fe^{2+}$)] content of coexisting olivine in peridotite samples. **d** Compositional variations of Cr# vs. $TiO_2$ (wt%) content of spinels in peridotite samples of the Munabulake ophiolite. Fields for passive margin, abyssal and forearc peridotites and degrees of melt extractions can be found in[31,55] and references therein. FMM Fertile midocean-ridge basalt (MORB)-type mantle, PM Primitive Mantle.

conclude to be have been formed during the commencement of subduction of the Proto-Tethys Ocean (Fig. S8). The reported ca. 517 Ma oceanic type adakite was generated by partial melting of oceanic crust in a newly formed subduction zone. Moreover, given the large-scale sinistral shearing of the south Altyn fault zone, we infer that the Munabulake ophiolite should be a member of the south Altyn ophiolite belt and was displaced to its current location by strike-slip motion along the fault system.

**Global plate re-organization at ca. 530–520 Ma during Gondwana assembly.** The Munabulake ophiolite dates the initiation of subduction of the Proto-Tethys Ocean in the Altyn segment, but the overall subduction initiation of this ocean is not well constrained and earlier inferred individual subduction zones have largely been treated in isolation (e.g.,)[13,14,39,61–63]. In addition, the ocean has been given a number of local names adjacent to the variety of continental and arc-related blocks in East Asia that are inferred to lie within the ocean. The early Paleozoic oceanic successions in these blocks are related to the evolution of the ocean and accretion of these blocks to the northern Gondwana margin (e.g.,)[13,14,52,62,63]. Nevertheless, in this study, a time-space

plot of early Paleozoic ophiolites and trench-arc assemblages across the East Asia blocks, along with related magmatic and metamorphic events, enables determination of the overall timing of initial oceanic subduction of the Proto-Tethys Ocean (Fig. 5). In particular, elsewhere across East Asia, ages of initiation of subduction are inferred from the oldest arc magmatism, which are in accordance with stratigraphy and ages of the ophiolites and metamorphic events (Fig. 5). We conclude that the main branch of the Proto-Tethys Ocean principally subducted northward and commenced at ca. 533 Ma in the West Kunlun segment[39,64], at ca. 525–520 Ma in the Altyn–Qaidam–Qilian segments[48,61]; (this study), and at ca. 515 Ma in the North Qinling segment (Fig. 6)[62]; Local and possibly isolated subduction zones in the southern Qaidam, Qiangtang and Indochina segments commenced at sometime around or before 535 and 490 Ma[13,14,65], but these dates are not well constrained. Therefore, overall timing of oceanic subduction initiation of the main Proto-Tethys Ocean youngs eastwards, commencing at ca. 533 Ma in the west segment and extending to ca. 515 Ma in the easternmost segment.

It is noteworthy that the timing of initial subduction of the Proto-Tethys Ocean coincides well with the timing of proposed slab roll-back of the Pacific Ocean in the southern Gondwana margin at

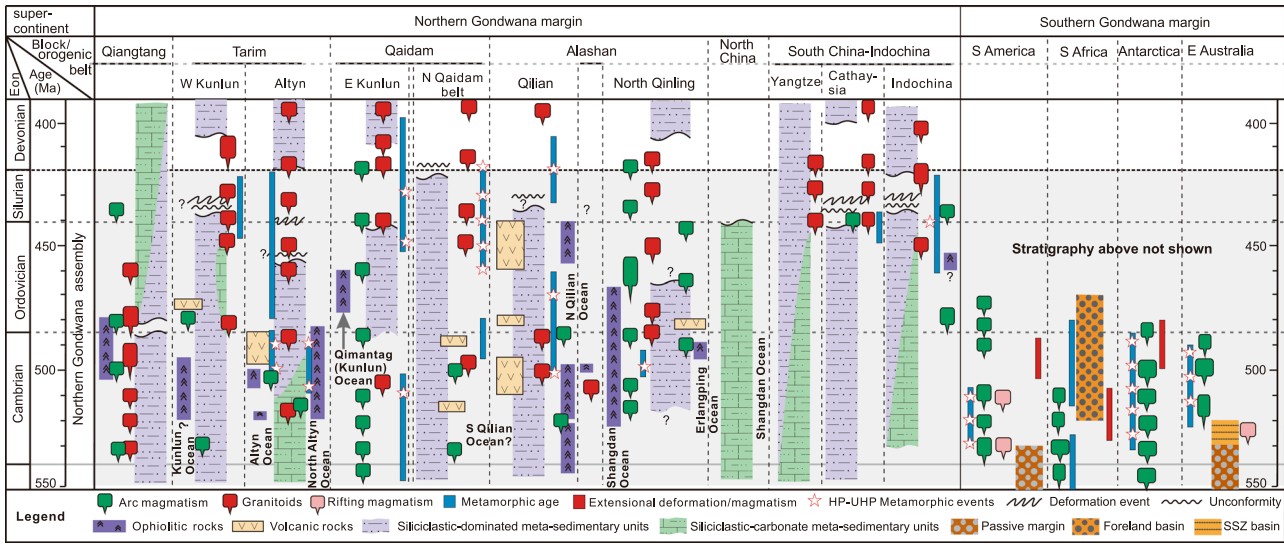

**Fig. 5 Time-space plot for the early Paleozoic sequences, ophiolites, stratigraphic sequences, and metamorphic events in Proto-Tethys and Pacific margins of Gondwana.** The complete list of references for age data points is given in the supporting information.

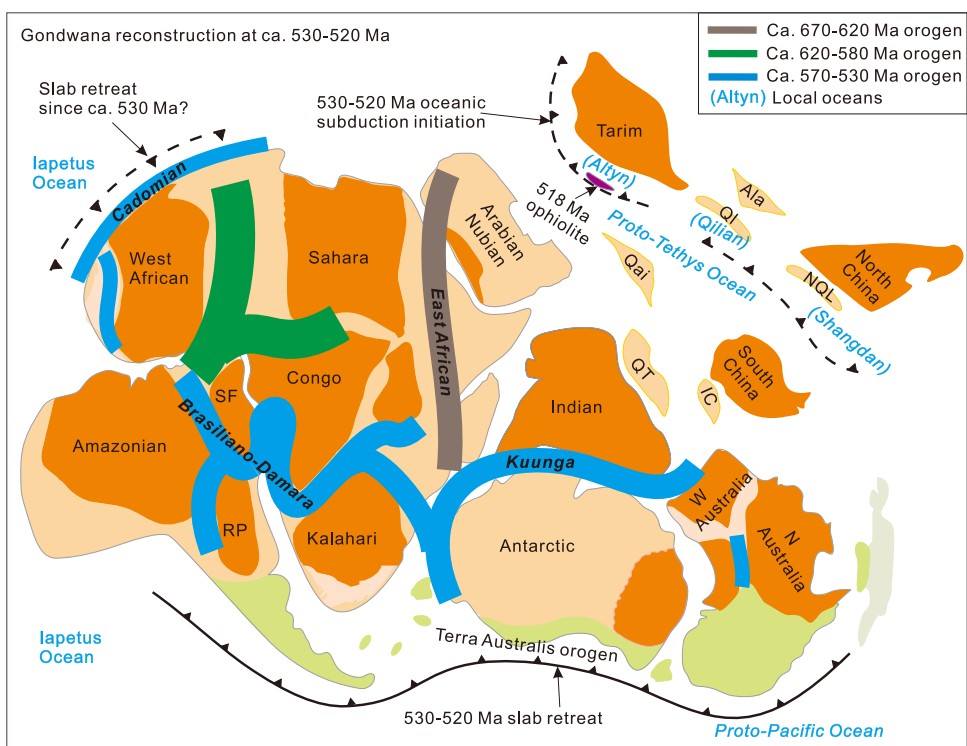

**Fig. 6 Reconstruction of Gondwana and subduction zones in its southern, western, and northern margins (revised based on)[13–16,63].** *Ala*, Alashan; *Ql*, Qilian; *Qai*, Qaidam; *QT*, Qiangtang; *NQL*, North Qinling; *IC*, Indochina; *SF*, São Francisco; *RP*, Río de La Plata.

530–520 Ma[13], and the change from oblique subduction to lithospheric extension along the West Gondwana margin at ca. 530 Ma[66,67] (Fig. 6). Thus, the tectonic regimes along the southern and western Gondwana margins are linked to that along the northern Gondwana, indicative of global plate re-organization at this time associated with the final collisional assembly of Gondwana. Moreover, modeling results indicate that Mariana-type subduction initiation is necessarily whole plate scale (>1000 km)[28,33], and intra-oceanic subduction initiation is more prevalent during times of supercontinent assembly[68], consistent with formation of the Munabulake ophiolite during final Gondwana assembly and a global tectonic re-organization.

**Mariana-type subduction initiation ophiolites in Earth history: establishment of modern plate tectonic regime and contemporary Earth.** Ophiolites formed during spontaneous (also referred to as 'vertical force or buoyancy driven') subduction initiation of a modern Mariana-type oceanic subduction zone include the 52 Ma IBM ophiolites and the Tonga ophiolite in west Pacific[27,30,31,33], 100–90 Ma Tethyan ophiolites[32,35], the ca. 335 Ma Paleo-Asian Ocean ophiolite[69], possible 490–485 Ma Appalachian-Caledonian ophiolites[70,71], the 518 Ma Munabulake ophiolite (Proto-Tethys ophiolite) in northern Gondwana margin in this study (Fig. 6), and the ca. 800-690 Ma Arabian-Nubian Shield ophiolites[72]. In addition, early Cambrian forearc ophiolites

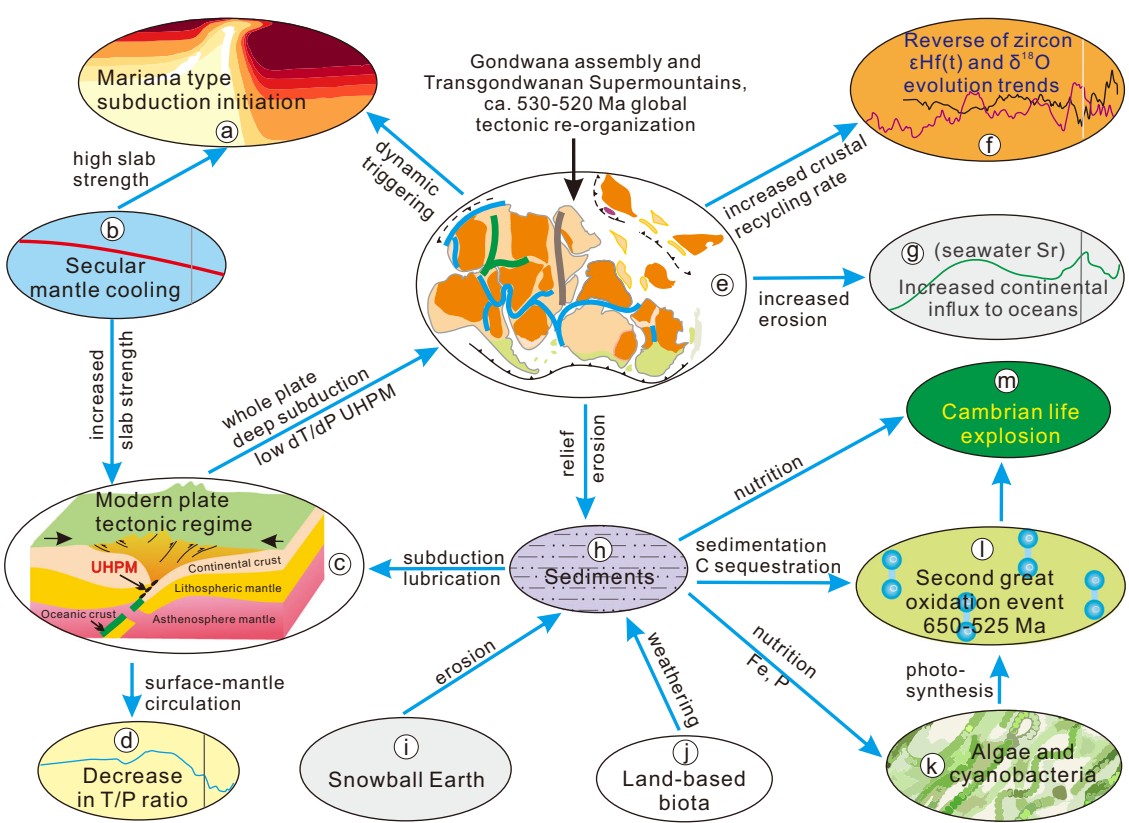

**Fig. 7 Conceptual model for the establishment of modern tectonic regime and it's links to tectonic events and surface environs.** Links and driving menchanism between varied tectonic and surfacial proxies are presented. (7a, 7b, 7d, 7f, 7g are after or based on[5,21,26,28,75] and references therein, respectively).

have also been previously identified[73], but Mariana-type ophiolite stratigraphy and chemical duality have not been observed from these ophiolites. The relatively restricted temporal distribution of one-sided modern Mariana-type oceanic subduction[2,28,33], essentially to the Phanerozoic, implies the establishment of a similar plate tectonic regime throughout this timeframe. The high strength of subducting oceanic plate required for the initiation of Mariana-type ophiolites, was likely controlled by progressive mantle cooling[5]. High slab strength, along with comparable ophiolite characteristics and scenarios of simultaneous oceanic subduction initiation since the Neoproterozoic-early Cambrian, coincides well with the inferred transition from an earlier tectonic regime to the modern plate tectonic regime in the Neoproterozoic-Cambrian[17,18,20,21].

Given the controlling factors of the development of plate tectonics, we conclude a new conceptual model for the development and establishment of the modern plate tectonic regime on Earth, which resulted in global tectonic re-organization and Mariana-type subduction initiation (Fig. 7). Gondwana assembly resulted in the establishment of a suite of super-mountains with a total length of 8000 km, in or close to the monsoon belt that were unprecedentedly high, with mountain roots exhuming UHP metamorphic rocks[1,17–20], leading to exceptionally high erosion rates[23,24]. The super-mountains occurred as result of deep subduction type modern plate tectonic regime that initiated and developed at this age period, but also contributed to high sediment flux to trenches with inferred lubrication of convergent plate interfaces which contributed to sustained plate tectonics, high global subduction activity, and significant rise of passive margins[6,74]. The increased erosion is well recorded by the increased continental derived component in seawater Sr (Fig. 1)[75]. The increased slab strength, along with global tectonic re-organization at ca. 530–520 Ma,

further resulted in the formation of the Mariana-type subduction initiation observed in this study. The increasing surface erosion rates supplied nutrition to oceans and enhanced C sequestration, responsible for the Ediacaran-Cambrian Great Oxidation Event and life explosion event[76,77], likely the key factors controlling overall Earth surficial evolution (Fig. 7).

The establishment of the modern plate tectonic regime corresponds with a global scale tectonic re-organization in the early Cambrian involving an increased sediment flux to the oceans, the global scale subduction of cold lithosphere into the deep mantle, and thus improved surface-mantle circulation efficiency, leading to cooling of Earth and the drop in mean thermobaric ratios T/P ratio at ca. 525 Ma (Fig. 1). The establishment of modern plate tectonic also led to gradually increased global scale crustal recycling as recorded by the detrital zircon initial mean εHf and δ18O trends that reached the lowest and highest values, respectively, in Earth history during the early Cambrian (Fig. 1), as intensified subduction improves crustal recycling rate.

## Data availability
All data generated during this study are included in the supporting information. Other additional information is available from the corresponding author upon reasonable request.

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

## Acknowledgements

This research is funded by the NSFC grants (41730213, 41972238, and 42072264), the State Key Laboratory of Continental Dynamics (201212000174) and the Hong Kong Research Grants Council General Research Fund (grants 17307918). P.A.C. acknowledges Australian Research Council grant FL160100168 for support. Drs. Zengchan Dong, Ningchao Zhou, and Shan Yu are thanked for their help during field work. The manuscript benefited from discussion with Drs. Jianhua Li, Yunying Zhang, Peiyuan Hu, Xiaojun Wang, Chao Wang, Xiangsong Wang and Mr. Jianjun Du. We also thank Drs. Jianfeng Gao, Ningchao Zhou, Zexian Cui and Xing Cui, and Mr. Liang Li for their helps with zircon isotopic and geochemical analyses.

## Author contributions

J.Y., P.A.C., and G.C.Z. designed the project and wrote the manuscript. J.Y., Y.G.H., Q.L., and P.W. conducted fieldwork. J.Y. and X.X. performed the analyses. All the authors contributed to the interpretation of the results and the revision of the manuscript.

## Competing interests

The authors declare no competing interests.
