## [Peer Review File · Nature Communications]

REVIEWER COMMENTS

Reviewer #1 (Remarks to the Author):

This paper presents critical evidence for the first appearance of Mariana-type ophiolites in the Earth history, that can be related to the establishment of modern-style plate tectonics regime on Earth. The transition to this regime was likely associated with Gondwana assembly between 800 and 550 Ma (e.g., Meert and Van Der Voo, 1997) and culminated with the proposed Mariana-type subduction appearance at around 533-515 Ma. This is very important new finding of broad interest. Indeed, it would be very useful to draw and discuss some kind of holistic conceptual scheme for the implied long-term (800-515 Ma) transition to modern plate tectonics. This scheme could possibly integrate this new finding with previously proposed concepts, such as for example influence of sediments delivery to trenches to speed up plate tectonics after the "boring billion" (Sobolev and Brown, 2019) that was possibly associated with a nearly static single-lid episode (Stern, 2020). It is in particular important to understand why Gondwana assembly did not associate with Mariana-type subduction but rather triggered it at a relatively late stage. It has been, for example, proposed that Mariana-type subduction initiation is vertically rather than horizontally forced that is reflected by specific magmatism (e.g., Maunder et al., 2020). It would be therefore useful to understand why and how favorable conditions for such subduction initiation style (e.g., Crameri et al., 2020) have been created for the first time by Gondwana assembly. Specific comments are given below.

Taras Gerya, Zurich, 17.03.2021

Specific comments

Line 380. «In addition, recent works also favor formation of Earth inner core in the Ediacaran (53), which might have feedback effects on plate tectonic regime.» What kind of feedback?

Line 382. We suggest that following the global plate re-organization at ca. 530-520 Ma during Gondwana assembly, a plate tectonic regime with products and slab strength similar to modern day Earth was achieved." It would be useful to discuss how Gondwana assembly proceeded without Mariana-type subduction. In general, it would be useful to draw some kind of conceptual sketch visualizing major stages in the long-term offset of global modern-style plate tectonics that culminated with appearance of Mariana-type subduction.

References

- Crameri, F., Magni, V., Domeier, M., Shephard, G.E., Chotalia, K., Cooper, G., Eakin, C.M., Grima, A.G., Gürer, D., Király, A., Mulyukova, E., Peters, K., Robert, B., Thielmann, M. (2020) A transdisciplinary and community-driven database to unravel subduction zone initiation. *Nature Communications*, 11, 1-14.
- Maunder, B., Prytulak, J., Goes, S., Reagan, M., 2020, Rapid subduction initiation and magmatism in the Western Pacific driven by internal vertical forces. *Nature Communications*, 11, Article number: 1874.
- Meert, G., Van Der Voo, R. (1997) The assembly of Gondwana 800-550 Ma. *Journal of Geodynamics*, 23, 223-235.
- Sobolev, S.V., and Brown, M., 2019, Surface erosion events controlled the evolution of plate tectonics on Earth: *Nature*, v. 570, p. 52-57.
- Stern, R.J., 2020. The Mesoproterozoic single-lid tectonic episode: Prelude to modern plate tectonics. *GSA Today*, v. 30, p. 4-10, <https://doi.org/10.1130/GSATG480A.1>.CC-BY-NC.

Reviewer #2 (Remarks to the Author):

Review for Yao et al., Nature Communications

Yao and coworkers presented a combined field and geochemical study of Munabulake ophiolite in northern Tibet, which may be by far the oldest preserved Mariana type oceanic subduction initiation ophiolite. These findings are important for tracing the evolution of plate tectonics and mantle cooling history in deep time. Overall, I find this paper well written. Below, I have a few comments and questions for the authors.

First of all, I would steer away from water in zircon. Hydrogen partitioning in zircon can be highly dependent on REE partitioning and may not reflect water contents in the melt (see De Hoog et al., 2014). Zircons recovered from Atlantic MOR gabbros contain water up to 1200 ppm, comparable to your zircons.

Lines 230-232. Peridotites spinels – Peridotite spinels. You are also missing the references here.

Line 269. “not shown”??? Why not show them?

Line 278-280. Many arc magmas are tholeiitic. In particular, most Mariana arc magmas are tholeiitic...

Line 380-381. I would remove the inner core stuff as it is distracting. If you want to keep it, you should briefly explain what the feedbacks are, or at least cite appropriate papers here.

Figure 4. Put error bars in both panels.

Figure 5. Maybe plot Mariana samples here for comparison?

Figure 9. I’m not sure if need this figure here. Your findings at this stage do not have any obvious implications for Earth’s system evolution in the Precambrian...You plotted seawater Sr isotope data here, but you never talked about seawater Sr in the text.

Reviewer #3 (Remarks to the Author):

Yao et al Mariana type oceanic subduction initiation Review by Brendan Murphy

This paper has four main goals:

1. To document a well exposed and strategically located ophiolite complex
2. To provide geochemical and isotopic data to deduce its tectonic setting, which support an evolution which includes Marianas-style subduction initiation
3. To demonstrate that this is the oldest known (ca. 518 Ma) example of Marianas-style subduction initiation.
4. To argue that these results imply comparable (subducted) slab strength and conditions similar to those that characterize modern plate tectonics.

I think the authors provide very persuasive arguments in favour of the first three goals. As currently written, many readers may think that the fourth goal is akin to “setting up a straw man” in that most

will assume modern plate tectonics since at least 2.0 Ga and so might feel underwhelmed by the conclusion. I understand what the authors are trying to get across, but their central point misfires slightly. To rectify this problem, I recommend a closer match between what is discussed in the Introduction and the Discussion (i.e. subduction initiation). Once this problem is rectified (and I think it can be rectified easily), I think the paper would make an admirable contribution to Nature Communications and I recommend publication after minor revisions.

I think some statements in the Introduction about the potential changes in the style of subduction over time would benefit. To that end, I suggest the authors consider moving Fig. 6 to become Fig 1, so to set up the context for how the data from the Munabulake provide insights into the problem. Fig 6 is a compilation that has (essentially) been previously published. The way the paper is currently set up, much of the statements about the transition to deep subduction (and hence "modern plate tectonics" could have been written without the Munabulake data (especially if HP/UHP metamorphism is a proxy for deep subduction)!!

Setting up the paper in this way would allow the reader to assess how the Munabulake data provide additional insights.

I also attach an annotated version of the manuscript, which for the most part is very well written. The authors should take care that my edits do not inadvertently change the meaning they intended.

REVIEWER COMMENTS

Reviewer #1 (Remarks to the Author):

This paper presents critical evidence for the first appearance of Mariana-type ophiolites in the Earth history, that can be related to the establishment of modern-style plate tectonics regime on Earth. The transition to this regime was likely associated with Gondwana assembly between 800 and 550 Ma (e.g., Meert and Van Der Voo, 1997) and culminated with the proposed Mariana-type subduction appearance at around 533-515 Ma. This is very important new finding of broad interest. Indeed, it would be very useful to draw and discuss some kind of holistic conceptual scheme for the implied long-term (800-515 Ma) transition to modern plate tectonics. This scheme could possibly integrate this new finding with previously proposed concepts, such as for example influence of sediments delivery to trenches to speed up plate tectonics after the “boring billion” (Sobolev and Brown, 2019) that was possibly associated with a nearly static single-lid episode (Stern, 2020). It is in particular important to understand why Gondwana assembly did not associate with Mariana-type subduction but rather triggered it at a relatively late stage. It has been, for example, proposed that Mariana-type subduction initiation is vertically rather than horizontally forced that is reflected by specific magmatism (e.g., Maunder et al., 2020). It would be therefore useful to understand why and how favorable conditions for such subduction initiation style (e.g., Crameri et al., 2020) have been created for the first time by Gondwana assembly. Specific comments are given below.

Taras Gerya, Zurich, 17.03.2021

Dear Professor Gerya, we very much appreciate your very insightful comments and suggestions, as well as the recommended papers, which are all very important in improving the manuscript, especially the recommend contribution of the role of sediments to development of plate tectonics. After careful reconsideration, we come up with a conceptual model for the establishment of modern plate tectonic regime and how it led to global tectonic re-organization and Mariana type subduction initiation at

ca. 530-520 Ma during Gondwana assembly.

The secular cooling of Earth's mantle is the primary factor for the development of modern plate tectonic regime, as researchers commonly agree, but the process was also kick-started and enhanced by the gradually increased surface erosion in the Neoproterozoic to early Paleozoic during Gondwana assembly, which resulted in a corresponding increase in sediment flux to trenches that served as lubricated the subduction zone interface and helped maintain stable subduction. This followed the possibly non-plate tectonic or nearly static 'boring billion' of Earth's middle age. The increased surface erosion could have been kick-started by the Snow ball Earth event (Sobolev and Brown, 2019), but was gradually dominated by erosion of the orogens formed during Gondwana assembly. The Gondwana assembly resulted in the establishment of a suite of super-mountains with a total length of over 8000 km, in or close to the monsoon belt, facilitating the generation and exhumation of UHP metamorphic rocks for the first time in Earth history. This led to exceptionally high erosion rates during Gondwana final assembly stage in the latest Neoproterozoic to early Cambrian period, which supplied sediments that enhanced and maintained the global scale modern type deep subduction.

The increased erosion is well recorded by the increased continental component in seawater Sr, which reached the highest values in Earth history in the early Paleozoic. This time frame also corresponds with a high crustal recycling rate as recorded by detrital zircon mean ϵ_{Hf} and $\delta^{18}\text{O}$ trends that reached the lowest and highest values, respectively, in Earth history during the early Cambrian.

The transition to modern plate tectonics was completed in the early Cambrian, which is whole plate scale and is marked in the geological record by global tectonic re-organization along the northern, southern, and western margins of Gondwana. We consider that Gondwana assembly related tectonic re-organization is also responsible for formation of the oldest Mariana type subduction initiation, which we observed in this study as intra-oceanic subduction initiation and this more prevalent during times of supercontinent assembly according to modelling results.

The established modern tectonic regime and global tectonic re-organization at ca.

530-520 Ma involved global scale subduction of cold lithosphere into deep mantle, leading to the sudden drop of T/P ratios at ca. 525 Ma. We also briefly concluded that the increasing surface erosion rates supplied nutrition to oceans and enhanced C sequestration, responsible for the Ediacaran-Cambrian Great Oxidation Event and life explosion event. Therefore, the development of modern tectonic regime is likely the key factors controlling overall Earth surficial evolution at this age period. In another words, the development and establishment of modern tectonic regime led to radical surficial climate and biosphere changes that established environments that characterize the contemporary Earth.

Specific comments

Line 380. «In addition, recent works also favor formation of Earth inner core in the Ediacaran (53), which might have feedback effects on plate tectonic regime.» What kind of feedback?

We deleted this sentence to maintain strict logic of the science of this manuscript. It is indeed too early to make assumption on how formation of Earth Inner core effected development of plate tectonics. The other two reviewers also suggest delete of this part.

Line 382. We suggest that following the global plate re-organization at ca. 530-520 Ma during Gondwana assembly, a plate tectonic regime with products and slab strength similar to modern day Earth was achieved.” It would be useful to discuss how Gondwana assembly proceeded without Mariana-type subduction. In general, it would be useful to draw some kind of conceptual sketch visualizing major stages in the long-term offset of global modern-style plate tectonics that culminated with appearance of Mariana-type subduction.

Numerical models indicate that the critical parameters that influence the position of subduction initiation are the lithospheric strength and the number of continental margins. The Mariana type spontaneous subduction initiation is driven by internal vertical forces and possibly occurred due to buoyancy heterogeneities along transform fault boundaries, but also requires high slab strength. This kind of subduction initiation is less common, while most subduction initiation is driven by horizontal

forces, so the Gondwana assembly likely proceeded without Mariana-type subduction.

We suggest that development of modern tectonic regime and final assembly of Gondwana led to global tectonic re-organization and Mariana type subduction initiation at ca. 530-520 Ma. This occurred as the primary condition to generate the Mariana-type subduction initiation, namely the high slab strength, was strong enough in the early Cambrian as mantle temperature gradually dropped. But the process is also triggered by global tectonic re-organization at ca. 530-520 Ma, as numerical modelling results indicate that intra-oceanic subduction initiation is more prevalent during times of supercontinent assembly.

References

Cramer, F., Magni, V., Domeier, M., Shephard, G.E., Chotalia, K., Cooper, G., Eakin, C.M., Grima, A.G., Gürer, D., Király, A., Mulyukova, E., Peters, K., Robert, B., Thielmann, M. (2020) A transdisciplinary and community-driven database to unravel subduction zone initiation. *Nature Communications*, 11, 1-14.

Maunder, B., Prytulak, J., Goes, S., Reagan, M., 2020, Rapid subduction initiation and magmatism in the Western Pacific driven by internal vertical forces. *Nature Communications*, 11, Article number: 1874.

Meert, G., Van Der Voo, R. (1997) The assembly of Gondwana 800-550 Ma. *Journal of Geodynamics*, 23, 223-235.

Sobolev, S.V., and Brown, M., 2019, Surface erosion events controlled the evolution of plate tectonics on Earth: *Nature*, v. 570, p. 52–57.

Stern, R.J., 2020. The Mesoproterozoic single-lid tectonic episode: Prelude to modern plate tectonics. *GSA Today*, v. 30, p. 4-10, <https://doi.org/10.1130/GSATG480A.1>.CC-BY-NC.

We appreciate these recommended references, which are all very useful in improving the manuscript and building a conceptual model for the development of modern tectonic regime, as well in building their link with global tectonic re-organization, and formation of the oldest Mariana type subduction initiation at ca. 530-520 Ma.

Reviewer #2 (Remarks to the Author):

Review for Yao et al., Nature Communications

Yao and coworkers presented a combined field and geochemical study of Munabulake ophiolite in northern Tibet, which may be by far the oldest preserved Mariana type oceanic subduction initiation ophiolite. These findings are important for tracing the evolution of plate tectonics and mantle cooling history in deep time. Overall, I find this paper well written. Below, I have a few comments and questions for the authors.

First of all, I would steer away from water in zircon. Hydrogen partitioning in zircon can be highly dependent on REE partitioning and may not reflect water contents in the melt (see De Hoog et al., 2014). Zircons recovered from Atlantic MOR gabbros contain water up to 1200 ppm, comparable to your zircons.

Thank you. We agree, the potential of hydrogen in zircon to interpret geological process needs more work, and we also want to be very careful with it, so we further revised and simplified the discussion.

There is debate over whether hydrogen partition in zircon is dependent on REE partition or the contrary, as there are very limited studies after De Hoog et al. (2014). A very recent publication by Meng and Xia et al. (2021) in *Science China: Earth Sciences* suggest that REE does not control hydrogen partitioning. Based on a series of laboratory studies, they proposed that water-in-zircon has the potential to probe water content of a magma.

The relations between water and REE in zircon are however, likely to be complex.

We would like to very cautiously mention that statistically, water contents of zircons from this study, with two populations at 100-320 ppm and 480-630 ppm, partially resemble those from the Atlantic MORB with one population at 100-300 ppm (De Hoog et al., 2014). Two zircons from this study also have high water contents, comparable a few that were also observed from the Atlantic MORB. In the revised manuscript, we have tried not to over-interpret the data and suggest the population at 480-630 ppm possibly recorded late-stage evolution from MORB to SSZ.

Moreover, we also genuinely hope that the data presented here can be useful to the geoscience world for water-in-zircon and subduction initiation study, as Mariana type

subduction initiation ophiolites are very rarely preserved, especially old ones such as reported here.

Lines 230-232. Peridotites spinels – Peridotite spinels. You are also missing the references here.

Revised as suggested, and reference to Yang et al. 2021. *Nature Reviews Earth & Environment* has been added to here.

Line 269. “not shown”??? Why not show them?

Thank you. We added a new related figure to the supporting information.

Line 278-280. Many arc magmas are tholeiitic. In particular, most Mariana arc magmas are tholeiitic...

Yes, indeed, most proto arc magmatism is tholeiitic. The tholeiitic part in here has been deleted. Sorry for the misunderstanding.

Line 380-381. I would remove the inner core stuff as it is distracting. If you want to keep it, you should briefly explain what the feedbacks are, or at least cite appropriate papers here.

Yes, we agree. The feedback effects of possible formation of the Earth inner core on plate tectonics seems indeed far-reaching. There is no data and observations to support a direct link. This discussion has been deleted to avoid over-interpretation of data and observations. Thanks again.

Figure 4. Put error bars in both panels.

Done as suggested.

Figure 5. Maybe plot Mariana samples here for comparison?

Thanks, we added new figure of Mariana subduction initiation ophiolite and proto-arc samples for comparison, which is included in the supporting information. It is indeed more convenient to compare the data patterns of this study with those of typical Mariana subduction zone.

Figure 9. I'm not sure if need this figure here. Your findings at this stage do not have any obvious implications for Earth's system evolution in the Precambrian...You plotted seawater Sr isotope data here, but you never talked about seawater Sr in the

text.

Thank you for the very useful comment. We largely expanded the discussion on the primary factors that contributed to initiation and development of modern plate regime in the Neoproterozoic-Cambrian, as well as the following plate tectonic re-organization and formation of the oldest Mariana type subduction initiation at ca. 530-520 Ma. We also build up a holistic conceptual scheme for the long-term transition to modern plate tectonics, which occurred as a combined effect of mantle cooling and increased surface erosion. We conclude that the modern plate tectonic regime was established during the final Gondwana assembly. In the revised version, the seawater Sr isotope is an important Earth evolution proxy that is indicative of increased surface erosion. We also introduced mean detrital zircon ϵ_{Hf} and $\delta^{18}\text{O}$ evolution trends to highlight increased recycling rates during development of modern tectonic regime. Please see text and reply to comments above.

Reviewer #3 (Remarks to the Author):

Yao et al Mariana type oceanic subduction initiation Review by Brendan Murphy

This paper has four main goals:

1. To document a well exposed and strategically located ophiolite complex
2. To provide geochemical and isotopic data to deduce its tectonic setting, which support an evolution which includes Marianas-style subduction initiation
3. To demonstrate that this is the oldest known (ca. 518 Ma) example of Marianas-style subduction initiation.
4. To argue that these results imply comparable (subducted) slab strength and conditions similar to those that characterize modern plate tectonics.

I think the authors provide very persuasive arguments in favour of the first three goals. As currently written, many readers may think that the fourth goal is akin to “setting up a straw man” in that most will assume modern plate tectonics since at least 2.0 Ga and so might feel underwhelmed by the conclusion. I understand what the authors are trying to get across, but their central point misfires slightly. To rectify this problem, I

recommend a closer match between what is discussed in the Introduction and the Discussion (i.e. subduction initiation). Once this problem is rectified (and I think it can be rectified easily), I think the paper would make an admirable contribution to Nature Communications and I recommend publication after minor revisions.

I think some statements in the Introduction about the potential changes in the style of subduction over time would benefit. To that end, I suggest the authors consider moving Fig. 6 to become Fig 1, so to set up the context for how the data from the Munabulake provide insights into the problem. Fig 6 is a compilation that has (essentially) been previously published. The way the paper is currently set up, much of the statements about the transition to deep subduction (and hence “modern plate tectonics” could have been written without the Munabulake data (especially if HP/UHP metamorphism is a proxy for deep subduction)!!

Setting up the paper in this way would allow the reader to assess how the Munabulake data provide additional insights.

I also attach an annotated version of the manuscript, which for the most part is very well written. The authors should take care that my edits do not inadvertently change the meaning they intended.

Dear Professor Murphy, we very much value your constructive comments. After careful reconsideration, we agree that the original manuscript has some shortcomings as to the link between the development of modern tectonic regime and the oldest Marian type subduction initiation, as well as the global tectonic re-organization at ca. 530-520 Ma. So, we have put considerable effort into building these links in the revised manuscript.

The introduction has also been considerably improved. The key factors, including slab strength controlled by mantle secular cooling and effects of sediment erosion, has been added to this part, along with potential changes in the style of plate tectonics over time. We also added potential changes of Earth climate and surficial proxies to the introduction, which are concluded to be controlled by the development of plate tectonics.

It is also a very good idea to move figure 9 (we believe you meant figure 9, instead of figure 6), which includes development of tectonic regimes and supercontinents over time, along with various Earth evolution proxies, to the beginning of the manuscript as figure 1.

In the revised manuscript, we largely improved the discussion on the link between initiation and establishment of modern plate tectonic regime in the Neoproterozoic-Cambrian with plate tectonic re-organization and formation of the oldest Mariana type subduction initiation at ca. 530-520 Ma during Gondwana assembly. We also established a conceptual scheme for the initiation and establishment of modern plate tectonics, which occurred as a combined effect of mantle cooling and increased surface erosion. The process was kick-started and enhanced by the gradually increased surface erosion in the Neoproterozoic to early Paleozoic during Gondwana assembly. This increased sediment flux helped lubricate the subduction interface and increase subduction velocity. The Gondwana assembly related super-mountains, which had a total length of 8000 km, straddled the monsoon belt, leading to exceptionally high erosion rates.

The increased erosion is well recorded by increased continental component in seawater Sr. The increased erosion rates, in combination with mantle cooling, also contributed to increased subduction activity, along with high crustal recycling rate as recorded by detrital zircon mean ϵ_{Hf} and $\delta^{18}\text{O}$ trends that reached the lowest and highest values in Earth history during the early Cambrian.

We would also like to thank you for very carefully correcting the manuscript, and the manuscript has been revised as according to your attached annotated version of the manuscript.

Specific comments in response to your annotated pdf file.

1. The Jixian and Changcheng systems are older stratigraphic terms that used to be used in China. We deleted them to avoid misunderstanding, and also because the manuscript is attended for international readers that are interested in global plate

tectonic regime and Earth evolution.

2. We deleted the parts that argue for layered marble within the ophiolite indicative of deep-sea environment, given that the marbles, though may indicate a marine environment, but cannot be directly dated and may not be of deep sea facies. The metamorphosed chert within the ophiolite would be sufficient evidence of formation in a deep-sea environment.

3. Diagram of element ratios that indicates hydrous fluids alteration has also added to the supporting information.

4. We deleted the sentence of the possible age of the formation of Earth Inner core to maintain strict logic of the science of this manuscript. It is indeed too early to make assumption on how formation of Earth Inner core effected development of plate tectonics. The other two reviewers also suggest delete of this part.

5. The water-in-zircon content is not inclusion, but water in mineral structure.

6. By deep subduction, we also did further investigation, and confirmed that slab can be subducted to a depth of 300 km before exhumation in the early Cambrian, which is likely a result of secular of mantle cooling and establishment of modern tectonic regime.

In addition, we have also revised the title of the manuscript to better highlight the contribution of this manuscript, as this study constrains the establishment, rather than initiation, of modern plate tectonic regime. The Mariana type subduction initiation and global tectonic re-organization are also important contributions of this manuscript.

The original Figure 3 of age data and figure 4 of REE and trace element patters were moved to the supporting information to avoid too many figures in the main manuscript.

A figure showing the conceptual establishment of modern plate tectonic regime, and links with global tectonic re-organization and Mariana type subduction at ca. 530-520

Ma, and changes in climate and Earth surficial proxies, has also been added to the manuscript. We believe it better highlights the contribution of this paper and also will help the readers to grab the idea of this contribution.

REVIEWERS' COMMENTS

Reviewer #1 (Remarks to the Author):

The Authors did careful revisions that properly addressed reviewers' comments.

The paper is in a good shape and I recommend it for publication.

Taras Gerya, Zurich, 24.05.2021

Reviewer #2 (Remarks to the Author):

The authors addressed all of my comments.

Reviewer #3 (Remarks to the Author):

The authors have done a conscientious job in addressing the issues I raised as well as issues raised by other reviewers. I recommend acceptance.

Brendan Murphy

REVIEWERS' COMMENTS

Reviewer #1 (Remarks to the Author):

The Authors did careful revisions that properly addressed reviewers' comments.

The paper is in a good shape and I recommend it for publication.

Taras Gerya, Zurich, 24.05.2021

Dear Prof. Taras Gerya, thank you for approving our work. This paper greatly benefited from your earlier insightful and constructive comments. We very much appreciate it.

Reviewer #2 (Remarks to the Author):

The authors addressed all of my comments.

Thank you for the kind remarks and approving our revision work. Your comments and suggestions contributed a lot to improving of our work.

Reviewer #3 (Remarks to the Author):

The authors have done a conscientious job in addressing the issues I raised as well as issues raised by other reviewers. I recommend acceptance.

Brendan Murphy

Dear Prof. Brendan Murphy, thanks a lot for approving our revision work. Your earlier comments helped us to considerably improve our work.